# Sequencing Reveals miRNAs Enriched in the Developing Mouse Enteric Nervous System

**DOI:** 10.3390/ncrna10010001

**Published:** 2023-12-22

**Authors:** Christopher Pai, Rajarshi Sengupta, Robert O. Heuckeroth

**Affiliations:** 1The Children’s Hospital of Philadelphia Research Institute, Philadelphia, PA 19104, USA; cpai@pennmedicine.upenn.edu; 2Department of Pediatrics, The Perelman School of Medicine, University of Pennsylvania, Philadelphia, PA 19104, USA; 3American Association for Cancer Research, Philadelphia, PA 19106, USA; rajarshi.sengupta@aacr.org

**Keywords:** enteric nervous system, Hirschsprung disease, miR-9, miR-27b, miR-124, miR-137, miR-488, smRNA-seq

## Abstract

The enteric nervous system (ENS) is an essential network of neurons and glia in the bowel wall. Defects in ENS development can result in Hirschsprung disease (HSCR), a life-threatening condition characterized by severe constipation, abdominal distention, bilious vomiting, and failure to thrive. A growing body of literature connects HSCR to alterations in miRNA expression, but there are limited data on the normal miRNA landscape in the developing ENS. We sequenced small RNAs (smRNA-seq) and messenger RNAs (mRNA-seq) from ENS precursor cells of mid-gestation *Ednrb-EGFP* mice and compared them to aggregated RNA from all other cells in the developing bowel. Our smRNA-seq results identified 73 miRNAs that were significantly enriched and highly expressed in the developing ENS, with miR-9, miR-27b, miR-124, miR-137, and miR-488 as our top 5 miRNAs that are conserved in humans. However, contrary to prior reports, our follow-up analyses of miR-137 showed that loss of *Mir137* in *Nestin-cre*, *Wnt1-cre*, *Sox10-cre*, or *Baf53b-cre* lineage cells had no effect on mouse survival or ENS development. Our data provide important context for future studies of miRNAs in HSCR and other ENS diseases and highlight open questions about facility-specific factors in development.

## 1. Introduction

The enteric nervous system (ENS) is a network of neurons and glia in the bowel wall that controls bowel motility [1], epithelial cell function [2], and blood flow to the bowel [3,4]. The ENS primarily derives from the vagal neural crest [5], a highly migratory, multipotent cell population. At embryonic day 9.5 (E9.5) in mice, pre-enteric neural crest–derived cells invade the foregut then migrate and proliferate along the fetal bowel until, at E13.5, the bowel is fully colonized [6]. The enteric neural crest–derived cells (ENCDCs) then separate into two layers of ganglia, the myenteric plexus and the submucosal plexus [7]. Select ENCDCs stop migrating and differentiate into electrically active neurons by E11.5 [8], though the ENS continues to develop and mature in utero and postnatally [9,10]. The resulting diverse and complex ENS network is essential for survival.

One of the most dramatic failures in ENS development is evidenced in the congenital disorder Hirschsprung disease (HSCR). In HSCR, ENCDCs fail to fully colonize the distal bowel during the first trimester of pregnancy, leading to a lack of enteric neurons at the end of the bowel. The region that lacks the ENS tonically contracts, preventing the passage of intestinal contents. HSCR is characterized by severe constipation, abdominal distention, bilious vomiting, and failure to thrive. Current HSCR treatment is surgical removal of the affected bowel, but bowel inflammation and dysmotility frequently persist after this surgery [11]. A growing body of literature has linked HSCR to alterations in miRNA expression, suggesting a potential avenue for future research.

In the ENS, miRNAs are essential for progenitor differentiation and cell survival. This was demonstrated using *Wnt1-cre; Dicer^fl/fl^* mice, where deletion of *Dicer* (an essential miRNA processing enzyme) blocks miRNA biogenesis. Although ENCDCs in *Wnt1-cre; Dicer^fl/fl^* mice colonize the bowel normally by E13, *Wnt1-cre; Dicer^fl/fl^* ENCDCs almost completely disappear by E17 [12]. In contrast, in WT mice, ENCDCs complete bowel colonization by E13.5, separate into nascent myenteric and submucosal plexuses, and begin differentiating into mature neurons between E13 and E17. While the ENCDCs in *Wnt1-cre; Dicer^fl/fl^* mice are likely lost as a result of mass apoptosis [13], these experiments suggest a window of time when miRNAs are essential for ENS development.

In this study, we hypothesized that one or more miRNAs expressed at E13.5 are critical for ENS development and attempted to identify them. We report the results of small RNA sequencing (smRNA-seq) of E13.5 ENCDCs. We found 73 miRNAs highly expressed and significantly enriched in ENCDCs compared to 141 miRNAs enriched in other cells in the bowel. We identified miR-137 as a potential miRNA of interest based on prior literature about its influence in neuronal differentiation in the central nervous system. We evaluated survival of *Mir137* conditional knockout mice using four Cre-driver lines. Surprisingly, we found a lack of mouse mortality in any conditional knockout line, despite widespread lineage labeling and in contrast to prior studies.

## 2. Results

### 2.1. Paired smRNA-seq and mRNA-seq of Fetal ENCDCs Identified miRNAs with Potential Roles in Development

We isolated GFP^+^ ENCDCs and GFP^−^ gut cells from the bowels of E13.5 *Ednrb-EGFP* mice by fluorescence-activated cell sorting. *Ednrb-EGFP* mice produce GFP in all ENS cells at E13.5 [14]. Bulk RNA from GFP^+^ and GFP^−^ cells was separately sequenced to identify small RNA (smRNA-seq, filtered for miRNAs before the final analysis) regulators and messenger RNA (mRNA-seq) that could be regulated by miRNA (Figure 1A). We compared GFP^+^ and GFP^−^ smRNA-seq data to identify miRNAs enriched in ENCDCs (Figure 1B, Appendix A). We selected miRNAs that had an average expression value of at least 4 CPM (assuming miRNAs must clear a minimum threshold to effectively regulate mRNA) and were specifically enriched in ENCDCs, narrowing our results to 73 miRNAs. The top six high-expressing significantly ENCDC-enriched miRNAs were miR-9, -6540, -27b, -137, -124, and -488 (Table 1). As miR-6540 is only present in mice, additional analyses of miR-6540 were not pursued.

To identify potential mRNA targets for the five top miRNAs of interest, we used TargetScanMouse v8.0 [15,16] and compared the predicted mRNA targets to the mRNAs present in GFP^+^ ENCDCs based on our mRNA-seq data (Appendix A). The resulting list of present predicted targets was then processed with the PANTHER Classification System to find biological processes each miRNA could regulate (as identified by the Gene Ontology (GO) Consortium) [17,18,19] (Figure 1C). The GO terms identified consistently involved development, neurons, and synapses, suggesting our cell isolation, sequencing, and data analysis pipelines correctly identified miRNAs enriched in ENCDCs at a critical juncture in development. Interestingly, of the 17,229 genes identified by mRNA-seq, 13,308 genes (77.2%) were not targeted by any of the 5 miRNAs of interest, and only 3 genes (*Nfia*, *Nfib*, and *Zbtb20*) were predicted to be targeted by all 5 miRNAs (Appendix A).

### 2.2. Paired smRNA-seq and mRNA-seq Data from Normal Bowels Are Insufficient to Predict Affected Biological Processes

To identify potential biological processes affected by miRNA targeting, we tested two pipelines for synthesizing smRNA-seq and mRNA-seq data. The first pipeline weighted predicted TargetScan 7.2 scores by the average expression of their respective miRNAs to create a predicted targeting strength score. The *t*-statistics of the mRNA-seq differential expression analysis were then multiplied by their predicted targeting strength scores with the intent to elevate low-enrichment, highly targeted mRNAs to significance. Significant *t*-statistics were identified via a permutation analysis, ultimately identifying 31 genes. A secondary analysis using randomized mRNA-seq read counts for differential expression and pipeline testing identified 25 significant genes, for a signal-to-noise ratio of 1.24. We deemed this ratio insufficient to be considered reliable. When the analysis was limited to targeting by miR-137 alone rather than all 73 ENCDC-enriched and highly expressed miRNAs, 18 significant genes were identified with only 2 present in the randomized analysis, for a more believable signal-to-noise ratio of 9. Of these 18 genes, 4 were enriched in ENCDCs (*Slc25a5*, *Snrk*, *Tcf4*, and *Neurod4*).

The high recovery of non-ENCDC-enriched genes by the first pipeline and its low signal-to-noise ratio when using all miRNAs led us to develop a second pipeline using solely ENCDC-derived sequencing data to evaluate miR-137 alone. After normalization, present genes were tested for expression >0 by a *t*-test, using a 5% false discovery rate as a threshold for significance. The *t*-statistics were then divided by their TargetScan scores for miR-137; thus, the *t*-statistics of the potentially repressed genes were further lowered by their predicted strength of targeting by miR-137. Significantly low *t*-statistics were identified by bootstrapping, and genes that were not originally significantly expressed were filtered out of the dataset. Ultimately, this pipeline identified nine genes of interest (*Msi1*, *Raver2*, *Lmtk2*, *Kcna2*, *Asphd1*, *Neurod4*, *B3galt2*, *Prdm10*, and *Kcnab3*).

GO term analyses of the results of both pipelines failed to produce any significant results. We thus concluded that analysis of paired smRNA-seq and mRNA-seq data under normal conditions is insufficient for identifying biological processes that are highly influenced by miRNAs and thus cannot guide in vivo experimentation.

### 2.3. Selection of miR-137 as a Candidate Regulator of ENS Development

We performed literature searches for each candidate miRNA to determine which ones we should follow up with further experimentation. We found that miR-9 and miR-124 both had well-established roles in central nervous system biology [20,21,22,23] and were capable of directly transforming fibroblasts into neuronal progenitors [24,25]. Meanwhile, miR-27b is frequently studied in the context of the heart [26,27], and miR-488 is suggested to be associated with many disease and cancer states [28], but neither seemed closely tied to neuronal development. We decided to pursue further analyses of miR-137, as multiple reports suggest that miR-137 is involved in hippocampal neuronal development, though with differing indications for its specific role [29,30,31]. We also noted that neuron-targeted *Mir137* conditional knockout (cKO) mice all died by P30 [32], a common finding in mice with ENS defects, whereas dramatic loss of miR-9 or miR-124 is perinatal lethal [33,34]. Before proceeding, we validated that miR-137 was enriched in ENCDCs by RT-qPCR (Figure 1D).

### 2.4. Mir137 Is Dispensable in ENCDCs, Neuronal Progenitors, and All Adult Neurons

The *Mir137^fl^* allele developed by Cheng et al. [32] is a functional *Mir137* allele that is excised in cells expressing Cre recombinase, allowing for lineage-specific conditional deletion of *Mir137* using Cre driver lines. We used this allele to test the necessity of *Mir137* expression in four Cre driver lines: *Nestin-cre*, *Wnt1-cre*, *Sox10-cre*, and *Baf53b-cre*. These Cre driver lines have been shown to target neuron progenitors, pre-migratory neural crest cells, migratory neural crest cells, and all mature neurons, respectively, and should thus all target the ENS in addition to other cell types.

We frequently observed germline recombination in *Nestin-cre* and *Sox10-cre* progeny, consistent with previous reports [35]. We also recovered one instance of germline recombination in *Wnt1-cre* progeny and tracked it to one male (out of dozens used to maintain the line over generations), indicating that germline recombination is rare but still possible using this *Wnt1-cre* allele. Finally, we observed germline recombination in *Baf53b-cre* progeny consistently and exclusively when *Baf53b-cre^+^* males were used during mating. Thus, only *Baf53b-cre^+^* females were used to maintain experimental lines.

We genotyped over 80 mice for each Cre driver line from *cre^−^; Mir137^fl/+^* x *cre^+^; Mir137^fl/+^* matings and assessed the ratios of genotypes we recovered (Table 2, Table 3, Table 4 and Table 5). We confirmed that mouse sex and genotype did not interact across all four lines and found that none of the lines tested resulted in a significant loss of *cre^+^; Mir137^fl/fl^* cKO mice by P28 (Table 6), contrasting with original reports [32].

The survival results suggested that despite miR-137 being highly enriched in E13.5 ENCDCs, it was not essential for ENS development. To confirm this hypothesis, we harvested the bowels from an adult *Wnt1-cre^+^; Mir137^+/+^; Rosa26^LSL-TdTomato^* mouse and an adult *Wnt1-cre^+^; Mir137^fl/fl^; Rosa26^LSL-TdTomato^* mouse. We used *Rosa26^LSL-TdTomato^* so that all cells that had ever expressed *Wnt1-cre* (i.e., all cells that should lack *Mir137* expression in cKO mice) would contain a fluorescent reporter. We stained the peeled proximal small intestine (PSI), distal small intestine (DSI), and distal colon (DC) with ANNA1, a human anti-neuronal antibody, and antibodies raised against S100β, a calcium-binding protein present in a subset of enteric glia (Figure 2, Appendix A). Comparison of these bowels showed a similar apparent ENS structure in both genotypes with no selective loss of TdTomato^+^ cells and similar densities of neurons (ANNA1^+^/S100β^−^ cells), glia (ANNA1^−^/S100β^+^ cells), and double-positive (ANNA1^+^/S100β^+^) cells (Table 7), further indicating that miR-137 is not necessary for development or maintenance of enteric neurons or glia in the small bowel or the colon.

### 2.5. Nestin-cre Targets Most Enteric Neurons and Glia, along with Many Other Cells within the Bowel

All *Nestin-cre* cKO mice were originally reported to die by P30 [32], but in our facility, *Nestin-cre; Mir137^fl/fl^* mice consistently survived to this date and beyond (Table 2). Although *Nestin-cre* lineage cells have been implicated in ENS development [36], there are limited data on *Nestin-cre* as a method of targeting the ENS; therefore, we used *Nestin-cre; Rosa26^LSL-H2B-mCherry^* mice to evaluate *Nestin-cre* lineage labeling in the adult ENS and confirm that *Nestin-cre* should induce loss of *Mir137* in the ENS. We stained the peeled proximal small intestine (PSI), distal small intestine (DSI), and distal colon (DC) with ANNA1 and antibodies raised against S100β (Figure 3, Appendix A).

In the myenteric plexus (Figure 3A–C), we found that neurons were present at densities ranging from 203.9 to 294.4 neurons/mm^2^ depending on the region (Figure 3D), with *Nestin-cre* lineage markers labeling 77.6–93.4% of neurons (Figure 3G). Glia were present at densities ranging from 176.9 to 326.8 glia/mm^2^ (Figure 3E), with high lineage labeling (94.5% and 94.4%) in PSI and DSI but comparatively low labeling (75.1%) in DC (Figure 3G). We also recovered rare ANNA1/S100β double-positive cells with estimated densities of 0.7–4.2 cells/mm^2^ (Figure 3F), though we could not confidently assess how often they were lineage traced by *Nestin-cre*.

In the submucosal plexus (Figure 3H–J), neurons were present at densities ranging from 18.3 to 62.6 neurons/mm^2^ (Figure 3K) and were labeled 77.3–94.8% of the time (Figure 3N). Glia were present at densities ranging from 83.5 to 126.9 glia/mm^2^ (Figure 3L), with lineage markers in 79.8–90.5% of glia (Figure 3N). Meanwhile, ANNA1^+^/S100β^+^ cells were significantly more common in PSI at a density of 67.7 cells/mm^2^, compared to 23.6 cells/mm^2^ in DSI and 1.6 cells/mm^2^ in DC (Figure 3M). These cells were generally lineage traced by *Nestin-cre*, with labeling rates ranging from 86.8 to 94.6% (Figure 3N).

While evaluating the ENS, we noticed that many ANNA1/S100β unstained cells were also labeled with H2B-mCherry, indicating that they descended from *Nestin-cre*-expressing precursors. While some cells could potentially be unstained neurons or glia, many of these cells morphologically and spatially appeared to be visceral smooth muscle (Appendix A), vascular smooth muscle (Appendix A), or epithelial cells (Appendix A). Combined, these results indicate that *Nestin-cre; Mir137* cKO mice should lack miR-137 in the vast majority of the ENS along with lacking miR-137 in many other cell types.

## 3. Discussion

The ENS has been called the “second brain” in the bowel due to its complex development, diverse cell types, and wide range of functions [37,38]. The cells that make up the ENS arise initially from ENCDCs that proliferate and migrate along the bowel during fetal development. When ENCDCs fail to colonize the distal bowel, the child is born with HSCR. While the vast majority of people with HSCR have at least one known risk allele, direct causes of HSCR remain difficult to identify [39]. This, combined with the importance of environmental factors during ENS development [40,41], suggests that further research into programmatic effectors of development is necessary. miRNAs are well-positioned to perform this role but have been under-investigated in ENS development, in part because miRNA expression in the developing ENS has not previously been characterized.

Our sequencing data capture a snapshot of miRNA expression at E13.5 in the developing murine ENS and surrounding gut cells. At this time point, ENCDCs have finished colonizing the fetal bowel and are beginning to separate and differentiate into the presumptive myenteric and submucosal plexuses, with some cells already showing electrical activity. Of the 880 miRNAs recovered in our sequencing, we identified 73 miRNAs that are significantly enriched in ENCDCs and expressed at moderately high levels. The six most significantly differentially expressed genes in that ENCDC-enriched subset were miR-9, -6540, -27b, -137, -124, and -488. miR-6540 is not conserved in humans (and thus is not discussed further) but has the same seed sequence as miR-124, which may suggest similar functions in development.

miR-9 has been thoroughly studied during central nervous system development [20,21]. It reaches peak expression in mouse telencephalon at E13.5, where it controls neural progenitor proliferation and differentiation through targeting *Foxg1*, *Meis2*, *Gsh2*, and *Isl1* [34]. In the ENS, loss of miR-9 targeting sites in farnesyl-prelamin A results in esophageal achalasia [42], suggesting that miR-9 plays a critical role in maintaining adult ENS health. These results are narrowly relevant to progeria, however, so targeted studies examining the role of miR-9 in the ENS are necessary to investigate if there are additional roles.

miR-27b is frequently studied in the context of the heart, where it is suggested to mitigate atrial fibrillation and fibrosis but aggravate cardiac hypertrophy [26,27]. In the brain, miR-27b targets transcriptional repressors in the presynaptic transcriptome [43], with conflicting evidence about whether miR-27b is up- or downregulated in response to sevoflurane treatment (a model of neuroinflammation) [44,45]. There are no publications examining the role of miR-27b in the ENS or HSCR.

miR-124, like miR-9, is well-characterized in neurodevelopment [22,23]. When overexpressed in fibroblasts in conjunction with miR-9 and anti-apoptotic proteins, miR-124 induces transdifferentiation into neuronal precursors through widespread transcriptome regulation [24,25,46]. While miR-124 has been studied in HSCR, both miR-124 and its proposed target gene *SOX9* were upregulated in the stenotic bowel compared to the normal bowel from the same patient [47], indicating further research is required.

miR-488 is usually (but not always) downregulated in cancer, acute myocardial infarction, and other conditions [28]. In the brain, miR-488 plays an anti-apoptotic role and can limit neuronal apoptosis in epilepsy or ischemic stroke [48,49]. In the HSCR colon, however, miR-488 is upregulated, where it is suggested to target *DCX* to limit cell migration and proliferation [50].

Finally, miR-137 is another commonly studied miRNA in neurodevelopment [51]. Polymorphisms of *MIR137* are associated with both schizophrenia and autism spectrum disorder [52,53]. In embryonic hippocampal neuronal stem cells (NSCs), miR-137 reportedly targets *Kdm1a* to promote NSC migration and differentiation [29], while in adult hippocampal NSCs, miR-137 is proposed to target *Ezh2* and *Mib1* to inhibit NSC differentiation [30,31]. Germline *Mir137* KO and *Nestin-cre; Mir137* cKO mice have been reported to die by P30 [32], a phenotype common in HSCR mouse models. Prior to this study, however, there were no publications examining the role of miR-137 in the ENS or HSCR.

Only three genes are predicted to be regulated by all five of these miRNAs at once: the nuclear factor one genes *Nfia* and *Nfib* and the zinc finger protein *Zbtb20*. Conditional knockout of *Nfia* by *Wnt1-cre* results in the loss of A-fiber nociceptors in the dorsal root ganglia, which sense sharp pinprick sensations [54]. Germline knockout of *Nfib*, meanwhile, results in severe lung hypoplasia, and even heterozygosity of *Nfib* delays pulmonary differentiation [55]. Neither have been studied in the context of the bowel previously. *Zbtb20*, however, has been shown to mediate stress-induced visceral hypersensitivity in rats through its expression in dorsal root ganglia [56]. Mice lacking *Zbtb20* display multiple metabolic defects, many of which are independent of liver function [57]. All three merit further investigation during bowel development, especially given their central location in these miRNA regulatory networks.

Our attempts to synthesize our paired smRNA-seq and mRNA-seq data were unsuccessful at informing potential experiments to perform with *Mir137* cKO mice, as no biological processes were identified as significantly regulated by the identified targets. It is interesting to note, however, that both pipelines suggested that *Neurod4* may be a critical regulatory target of miR-137. *Neurod4*, also known as *Math3*, cooperates with *Ngn2* during development of the cerebral cortex [58]. Germline loss of *Neurod4* in mice causes severe growth and motor defects due to loss of granule cells in the cerebellum [59]. While *Neurod4* has not been studied in the context of the ENS, our work suggests it may merit further investigation with more targeted tools.

Interestingly, of the 37 miRNAs that we found that have been specifically studied in HSCR, only 83.8% (31/37) were sequenced in our dataset, and only 19.4% of those (6/31) were specifically enriched in ENCDC samples (Table 8). These discrepancies may be attributable to the mid-embryonic timepoint we sampled, our specific selection of ENCDCs for sequencing, or expression differences between the murine intestine and the human colon. We hope our sequencing data will inform future investigations into the role of miRNAs in HSCR and other ENS diseases.

We followed up our sequencing data by examining miR-137 across four Cre driver lines: *Nestin-cre* [98], the driver used by Cheng et al. [32] to study *Mir137* cKO mice; *Wnt1-cre* [99], which targets the pre-migratory neural crest and hindbrain; *Sox10-cre* [100], which targets the migratory neural crest and glia; and *Baf53b-cre* [101], which targets mature neurons and has been shown to label the myenteric plexus of the entire small intestine [102]. Surprisingly, we found no difference in survival resulting from conditional loss of *Mir137* in any Cre driver tested (Table 6). Preliminary analysis of ENS anatomy in a *Wnt1-cre^+^; Mir137^fl/fl^* mouse also appeared to be normal.

The contrast between previously published *Nestin-cre; Mir137* cKO results showing poor growth and death around or shortly after weaning [32] and our apparent lack of phenotypic differences between WT and cKO mice, coupled with the prominent ENCDC enrichment of miR-137 at E13.5, prompted additional investigation. While *Nestin-cre* has been used to target the ENS for studies previously [103], no study thoroughly characterized *Nestin-cre* lineage labeling of the adult ENS. Thus, we evaluated the number of ANNA1- and S100β-labeled cells that descended from *Nestin-cre* lineage cells. We found that the adult ENS is largely (but not completely) labeled by *Nestin-cre*-induced reporter expression. Additionally, while *Nestin-cre* was initially developed to target neurons and selectively constructed using the neuronal enhancer of the *Nestin* gene [98], we noted that many cell types beyond neurons and glia were labeled in *Nestin-cre; Rosa26^LSL-H2B-mCherry^* bowels.

Given the many non-neuronal cell types *Nestin-cre* is expressed in, it is worth considering how loss of *Mir137* in *Nestin-cre*-targeted non-ENCDCs may influence ENS development, even recognizing that miR-137 is barely detectable in non-ENCDCs at E13.5. In the developing bowel, the gut mesenchyme expresses *Edn3* in progressively more distal regions, supporting ENCDC migration down the bowel [104]. Later, as ENCDCs organize into the myenteric and submucosal plexuses, *Shh* expression in the bowel epithelia provides critical organizational information [105]. Neither are directly targeted by miR-137, but both serve to demonstrate the importance of crosstalk between ENCDCs and non-ENCDCs during development. Unfortunately, no study to date has reported on when *Nestin-cre* begins Cre-dependent recombination during bowel development, in the ENS or otherwise, rendering it difficult to assess when and how loss of *Mir137* may affect ENS development despite its broader targeting.

While our *Nestin-cre; Mir137* cKO results were unexpected, the differences we found are not without precedent. Even though we acquired the same strains as originally published and made significant efforts to minimize background mixing, differences in both the microbiome [106] and environmental stress [107] can have dramatic effects on mouse phenotypes. Unfortunately, critical facility-specific factors that impact phenotype can be challenging to identify, so we cannot readily assess why our results differed from prior reports.

## 4. Materials and Methods

### 4.1. Animals

Mice were maintained in accordance with a Children’s Hospital of Philadelphia (CHOP) IACUC-approved protocol (IAC 22-001041). *Mir137-flox* mice were a gift from Dr. Peng Jin [32] and maintained in the 129S6/SvEvTac background. *Ednrb-EGFP* (MMRC #066515-MU, RRID:MMRRC_066515-MU), *Nestin-cre* (JAX #3771, RRID:IMSR_JAX:003771), *Wnt1-cre* (JAX #3829, RRID:IMSR_JAX:003829), *Sox10-cre* (JAX #25807, RRID:IMSR_JAX:025807), *Baf53b-cre* (JAX #27826, RRID:IMSR_JAX:027826), *Rosa26^LSL-TdTomato^* (JAX #7909, RRID:IMSR_JAX:007909), and *Rosa26^LSL-H2B-mCherry^* (JAX #23139, RRID:IMSR_JAX:023139) mice were maintained in a C57Bl/6J background. Background mixing was limited when possible, using progeny from initial F1 crosses to maintain experimental colonies except to rescue a collapsing line. Further animal and housing information is in Appendix A.

### 4.2. Mid-Gestation ENCDC Isolation

Heterozygous *Ednrb-EGFP* mice and wild-type C57BL/6J mice were housed together and checked every morning for plugs, with the date of plug discovery considered E0.5. When the embryos reached E13.5, dams were euthanized via CO2, and the embryos were dissected in sterile L15 media (ThermoFisher #11415064, Waltham, MA USA) to isolate the bowel from the stomach to the anus. GFP^+^ bowels were identified with an inverted microscope (Zeiss Axio Observer.A1, Zeiss, Oberkochen, Germany; with attached X-Cite 120Q, Excelitas Technologies, Waltham, MA, USA) and incubated in sterile phosphate buffered saline (PBS; Mediatech #MT21-031-CM, Corning, NY, USA) with 0.2 mg/mL dispase (Gibco #17105041, Waltham, MA USA) and 0.2 mg/mL collagenase II (Sigma #C6885, Burlington, MA, USA) at 37 °C for 15 min. Loose cells were removed with PBS, then the embryonic bowel was fully dissociated by trituration with a P1000 pipette. The dissociated bowel was filtered with a 40 µm cell strainer (Corning #352340, Corning, NY, USA) before centrifuging at 293× *g* for 7 min. Pelleted cells were resuspended in 1 mL PBS with 2% heat-inactivated fetal bovine serum (Tissue Culture Biologicals #101, Tulare, CA, USA), 1 mM EDTA (Invitrogen #15575-038, Waltham, MA, USA), and 0.1% sodium azide (Sigma #S2002, Burlington, MA, USA) for fluorescence-activated cell sorting on a BD FACSJazz. After sorting to separate GFP^+^ ENCDCs from GFP^−^ gut cells (i.e., non-ENCDCs), samples were pelleted at 400× *g* for 10 min then resuspended in QIAzol (Qiagen #79306, Hilden, Germany) and stored at −80 °C until ready for RNA purification.

### 4.3. RNA Purification

RNA was purified from frozen samples in QIAzol using the Qiagen miRNeasy Mini Kit (Qiagen #217004, Hilden, Germany) in accordance with the manufacturer’s instructions. Briefly, the samples were thawed and allowed to solubilize for an additional 5 min after ice was no longer visible. The RNA fraction was then isolated via chloroform extraction. Ethanol was added to the clear hydrophilic fraction; then, the sample was loaded onto a spin column and washed with buffer RWT and RPE. The membrane was dried with an additional centrifugation step; then, the RNA was eluted into 50 µL of RNase-free water.

### 4.4. RNA Sequencing

Sequencing was performed by the Penn Genomics and Sequencing Core (PGSC; RRID: SCR_022382). Multiple RNA samples were pooled as necessary to obtain 4 replicates with 750 pg of RNA each. smRNA-seq libraries were prepared with the Illumina truSeq smRNA kit (Illumina #RS-200, San Diego, CA, USA), while mRNA-seq libraries were prepared with the Clontech SMARTer PCR cDNA Synthesis Kit (Clontech #634925, San Jose, CA, USA) followed by the Illumina NexteraXT kit (Illumina #FC-131, San Diego, CA, USA).

The sequencing reads produced by the PGSC were then aligned with STAR v2.4.2a [108] and normalized with PORT v0.8.5-beta [109]. Due to the short length of miRNAs, the additional arguments --outFilterScoreMinOverLread 0.3 and --outFilterMatchNminOverLread 0.3 were used to align the smRNA-seq data, thereby permitting short alignments. Minimum normalized read counts were used to analyze the mRNA-seq data, while the smRNA-seq data were analyzed using maximum normalized read counts due to gene duplication of many miRNA host genes (e.g., *Mir9-1hg*, *Mir9-2hg*, and *Mir9-3hg*). Differentially expressed genes were identified by processing read counts with the R voom package [110] then analyzing the output with the R limma package [111]. Finally, non-miRNA genes were filtered out of the smRNA-seq dataset, and duplicated miRNAs were consolidated by averaging their values.

### 4.5. RT-qPCR of miR-137

RNA samples were diluted to equivalent concentrations then prepared using the TaqMan Advanced miRNA cDNA Synthesis Kit (Applied Biosystems #A28007, Waltham, MA, USA). Briefly, RNA was polyadenylated then ligated to an adaptor before reverse transcription. The produced cDNA was then amplified to improve the detection of miRNA targets with low expression.

Quantitative polymerase chain reaction (qPCR) was performed using the TaqMan Fast Advanced Master Mix (Applied Biosystems #4444558, Waltham, MA, USA) and TaqMan Advanced miRNA Assays (Applied Biosystems #A25576, Waltham, MA, USA) for miR-137 (assay 477904_mir) and miR-106b (assay 478412_mir). miR-106b was chosen as a reference gene, as it has been shown to be more stably expressed than 5S rRNA or U6 snRNA during neuronal differentiation [112]. miR-137 expression was thus normalized to miR-106b expression, then ENCDC expression was normalized to gut expression. dCt(Gut)–dCt(ENCDC) (i.e., −ddCt) was chosen for the y-axis of Figure 1D so that increased miR-137 expression in ENCDCs produced a positive number.

### 4.6. Prediction of Critical mRNA Targets of Sequenced miRNA through Two Pipelines

The goal of both pipelines was to identify genes expressed in ENCDCs that could be repressed by miRNAs by enhancing their metrics of expression based on their predicted strength of repression. The first pipeline tested used the differential expression output for smRNA-seq and mRNA-seq. First, TargetScan 7.2 cumulative weighted context++ scores for every miRNA-mRNA pair were multiplied by the average expression value of each miRNA to incorporate the expected strength of targeting. Then, the *t*-statistic of each mRNA’s differential expression was multiplied by this targeting strength score. To evaluate weighted *t*-statistics for significance, 1000 permutations of this pipeline (using randomized TargetScan scores, miRNA expression levels, and mRNA *t*-statistics) were performed, and the boundaries of a two-tailed 5% analysis were identified. The signal-to-noise ratio was determined by randomizing the raw read counts of the mRNA-seq data and assessing these data via this differential expression and weighting pipeline.

The second pipeline tested only for predicted repression by miR-137. This pipeline first re-normalized the smRNA-seq and mRNA-seq data using only the ENCDC samples. Present miRNAs and mRNAs were identified by filtering for genes with an average of at least 3 transcripts across all samples and presence in at least 3 out of 4 samples, while significant expression was evaluated by a *t*-test for expression >0 (with a false discovery rate of <5% used as a threshold for significance). *t*-statistics were then divided by their TargetScan 7.2 cumulative weighted context++ score for miR-137; thus, the scores for potentially repressed genes were further lowered by their strength of targeting by miR-137. Significant *t*-statistics were again determined by a permutation analysis.

### 4.7. Survival Analysis

Mice from *cre^−^; Mir137^fl/+^* x *cre^+^; Mir137^fl/+^* matings were weaned at P28 and genotyped for wild-type, floxed, and deleted *Mir137* alleles using ear biopsies and previously published primers [32]. Mice with evidence of germline recombination were excluded from analysis, recognizing that each Cre driver line tested causes recombination in the skin (thus *Mir137^fl/-^* mice could not be differentiated from *Mir137^fl/fl^* mice). Whole litters were evaluated until at least 80 weanlings were genotyped, in order to produce a minimum expected value of 5 mice of each sex–genotype combination.

### 4.8. Adult Bowel Harvesting

Adult (P56-84) mice were euthanized by CO2 and dissected. The bowels were isolated from stomach to anus and roughly divided into the proximal small intestine, distal small intestine, and colon before being flushed with non-sterile PBS (ThermoFisher #21600069, Waltham, MA, USA). The flushed bowels were then opened lengthwise along the mesenteric border and pinned flat with stretching. The proximal and distal small intestine segments were trimmed to only the first and last 6 cm of pinned bowel, while the colon was divided into the proximal and middle/distal colon at the epithelial folds. The bowels were fixed with 4% paraformaldehyde (EMD Millipore CAS 30525-89-4, Burlington, MA, USA) in PBS for 25 min, rinsed with PBS, and stored overnight at 4 °C before peeling apart the muscularis and mucosal layers. The third centimeter of the proximal small intestine, the most distal centimeter of the distal small intestine, and the most distal centimeter of the colon were routinely used for further analyses. As the major duodenal papilla is typically found on the second centimeter of the proximal small intestine, the third centimeter was estimated to be the distal duodenum and/or proximal jejunum. The distal small intestine and distal colon segments corresponded to the distal ileum and rectum, respectively.

### 4.9. Immunohistochemistry

1 cm fixed bowel segments were blocked with 5% normal donkey serum (Jackson ImmunoResearch #017-000-021, RRID: AB_2337258, Ely, Cambridgeshire, UK) in PBS with 0.5% Triton X-100 (Sigma-Aldrich #T8787, Burlington, MA, USA) for 2 h at RT. The bowels were then stained with 1:20,000 ANNA1 (gift from Vanda Lennon, Mayo Clinic, Rochester, MN, USA, RRID: AB_2314657) and 1:200 rabbit anti-S100β (Abcam #ab52642, RRID: AB_882426, Cambridge, UK) in blocking buffer for 2 h at RT before being washed 3 times in PBS for at least 5 min per wash. The bowels were then stained with 1:400 AlexaFluor 647 Goat α-Human (Invitrogen #A21445, RRID: AB_2535862, Waltham, MA, USA) and 1:400 AlexaFluor 488 Goat α-Rabbit (Invitrogen #A11034, RRID: AB_2576217, Waltham, MA, USA) for 1 h at RT in the dark. Finally, the bowels were washed in the dark 3 times in PBS for at least 5 min per wash then mounted on a glass slide in 50/50 PBS/glycerol (Invitrogen #15514-011, Waltham, MA USA) with 0.05% sodium azide. Slides were sealed with clear nail polish (Sally Beauty #SBS-215100, Denton, TX, USA) and allowed to cure at RT overnight. Images were captured on a Zeiss LSM 710 confocal microscope with Zeiss ZEN 2.3 Sp1 FP3 (black) (Version 14.0.18.201; Zeiss, Oberkochen, Germany) software, a Plan-Apochromat 20×/0.8 air M27 objective lens, and slice interval of 1 µm. Images were taken from 8 unique fields of view (425.1 µm × 425.1 µm each), prioritizing the most proximal regions (if imaging the proximal small intestine) or the most distal regions (if imaging the distal small intestine or the colon).

Primary antibodies were considered trustworthy based on staining of *Wnt1-cre* lineage-traced cells, morphology and localization of staining, and comparison of images to published literature (including ANNA1^+^/S100β^+^ cells).

### 4.10. Cell Density Quantification

The bowel images were opened in ImageJ and quantified using the Cell Counter plugin. ANNA1^+^/S100β^−^ cells were considered neurons, while ANNA1^−^/S100β^+^ cells were considered glia. ANNA1^+^/S100β^+^ cells were also quantified but only referred to as “double-positive cells” for simplicity. Cell densities across all images of a given bowel and plexus region were averaged and reported as the density for that mouse (i.e., treated as an experimental N).

### 4.11. Statistics

A 6 × 2 contingency table of genotype × sex was constructed and evaluated using Fisher’s exact test to determine if genotype and sex were independent factors in determining survival to P28. As all experimental lines failed to disprove the null hypothesis of independence, sex was discarded as a differentiating factor, and the combined distribution of genotypes was compared to a null Mendelian distribution of genotypes (i.e., 12.5% for *cre^+^* and *cre^−^* with *Mir137^+/+^* and *Mir137^−/−^* and 25% for *cre^+^* and *cre^−^* with *Mir137^fl/+^*) using the chi-square test.

## 5. Conclusions

Our work provides a crucial perspective on the state of miRNA expression during ENS development as ENCDCs finish colonizing the bowel and begin differentiating into a mature neural network. We found 73 significantly ENCDC-enriched and highly expressed miRNAs and highlighted miR-9, -27b, -124, -137, and -488 as particularly of note. Our mRNA-seq dataset also provides important context for studying the potential effects of miRNA during ENS development. Our follow-up examination of *Mir137* revealed the importance of recognizing environmental factors in both mouse research and human disease.

## Figures and Tables

**Figure 1 ncrna-10-00001-f001:**
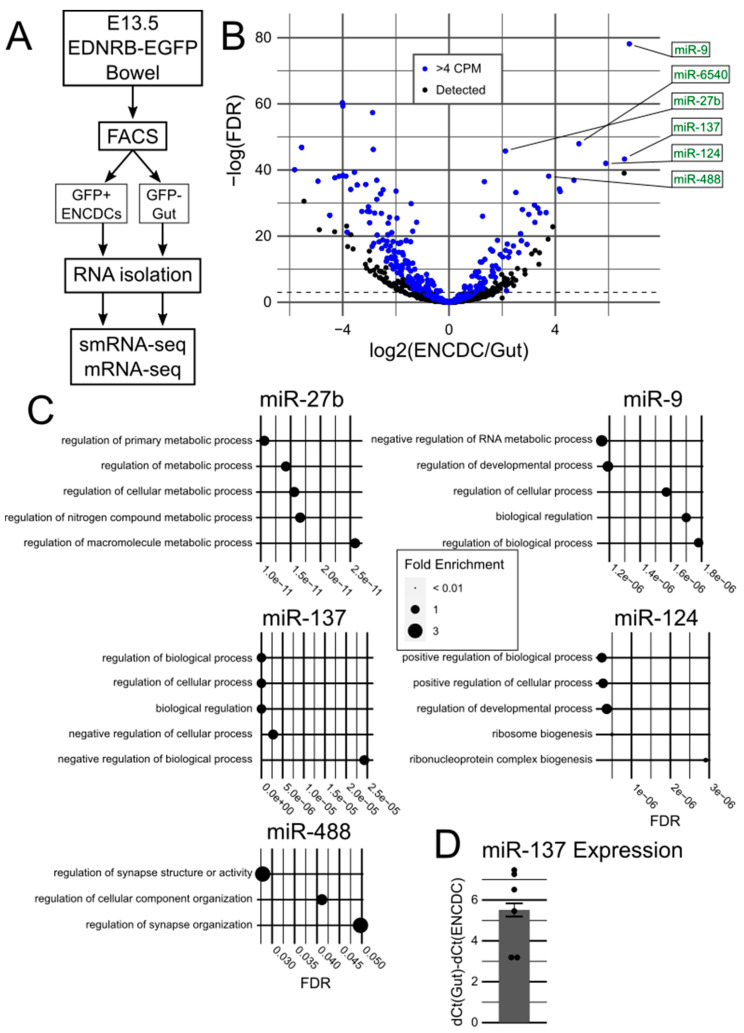
Analysis of ENCDC- and gut-paired smRNA-seq and mRNA-seq data. (**A**) Flowchart of analysis. E13.5 *Ednrb-EGFP* bowels were dissociated and sorted by FACS to separate GFP^+^ ENCDCs from GFP^−^ gut cells. RNA was isolated from each population then sequenced. (**B**) Volcano plot of miRNA results. Blue dots represent “highly expressed” miRNA, i.e., with >4 CPM detected. The top six most significant ENCDC-enriched miRNAs are labeled. (**C**) GO term enrichment analysis of the predicted mRNA targets of the top five ENCDC-enriched and evolutionarily conserved miRNAs. Dot size represents the fold enrichment of the term in the targeted gene pool compared to the reference pool of all present mRNAs. (**D**) RT-qPCR of miR-137 from six *Ednrb-EGFP* bowels comparing ENCDCs to other gut cells, demonstrating high enrichment of miR-137 in ENCDCs.

**Figure 2 ncrna-10-00001-f002:**
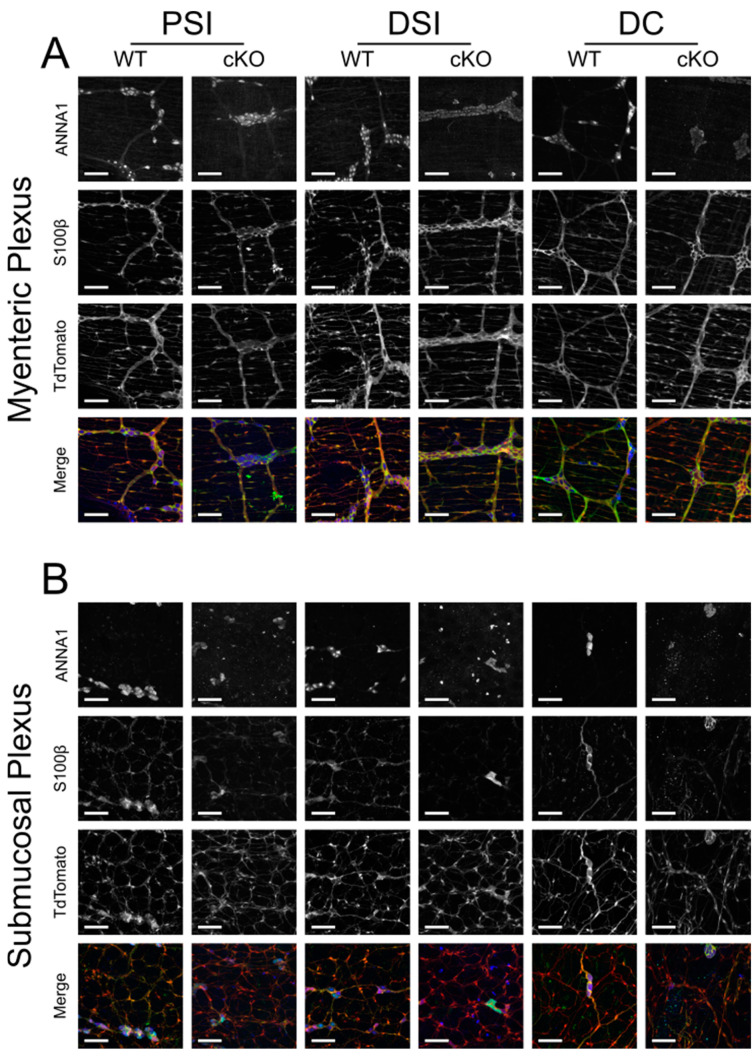
Loss of *Mir137* in *Wnt1-cre* lineage cells has no dramatic effect on the ENS. Split images of ANNA1 antibodies, S100β antibodies, and TdTomato protein in the (**A**) myenteric plexus and (**B**) submucosal plexus. Merged images are ANNA1 in blue, S100β in green, and TdTomato lineage tracing in red. Scale bars are 100 µm.

**Figure 3 ncrna-10-00001-f003:**
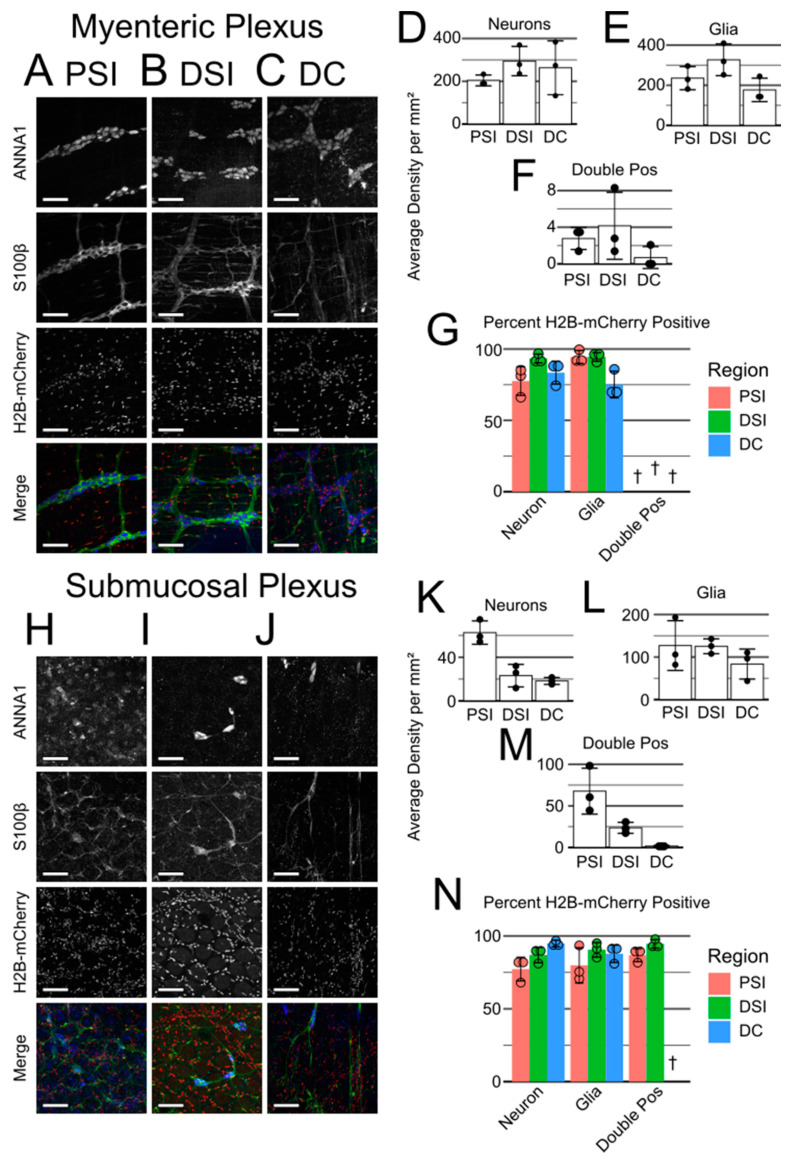
*Nestin-cre* lineage traces the majority of the ENS and many other cells. (**A**–**C**) Myenteric plexus of the PSI (**A**), DSI (**B**), and DC (**C**). (**D**–**F**) Graphs of the average density per mm^2^ of neurons (**D**), glia (**E**), and double-positive cells (**F**). (**G**) Graphs of the average percentage of neurons, glia, and double-positive cells that are lineage-traced by *Nestin-cre*. (**H**–**N**) As (**A**–**G**), but for the submucosal plexus. Merged images are ANNA1 in blue, S100β in green, and H2B-mCherry lineage tracing in red. Scale bars are 100 µm. † Percent lineage tracing cannot be confidently determined due to low cell density.

**Table 1 ncrna-10-00001-t001:** Top six most significantly differentially expressed miRNAs in enteric neural crest–derived cells (ENCDCs) over gut cells.

miRNA	log2(CPM)	log2(ENCDC/Gut)	FDR	Conserved?
miR-9	8.60	6.78	1.13 × 10^−34^	Yes
miR-6540	5.38	4.89	1.50 × 10^−21^	No
miR-27b	14.12	2.12	1.36 × 10^−20^	Yes
miR-137	2.31	6.60	1.57 × 10^−19^	Yes
miR-124	2.86	5.90	5.70 × 10^−19^	Yes
miR-488	6.50	3.75	2.71 × 10^−17^	Yes

**Table 2 ncrna-10-00001-t002:** Recovered weanlings from *Nestin-cre* litters.

	Male		Female		Total	
	Observed	Expected	Observed	Expected	Observed	Expected
*cre^+^ Mir137^+/+^*	6	5.5	5	5.5	11	10.25
*cre^+^ Mir137^fl/+^*	7	8.5	10	8.5	17	20.5
*cre^+^ Mir137^fl/fl^*	10	7	4	7	14	10.25
*cre^−^ Mir137^+/+^*	6	4.5	3	4.5	9	10.25
*cre^−^ Mir137^fl/+^*	9	8	7	8	16	20.5
*cre^−^ Mir137^fl/fl^*	6	7.5	9	7.5	15	10.25

**Table 3 ncrna-10-00001-t003:** Recovered weanlings from *Wnt1-cre* litters.

	Male		Female		Total	
	Observed	Expected	Observed	Expected	Observed	Expected
*cre^+^ Mir137^+/+^*	7	6	5	6	12	10.625
*cre^+^ Mir137^fl/+^*	9	9.5	10	9.5	19	21.25
*cre^+^ Mir137^fl/fl^*	5	4	3	4	8	10.625
*cre^−^ Mir137^+/+^*	2	3	4	3	6	10.625
*cre^−^ Mir137^fl/+^*	13	12.5	12	12.5	25	21.25
*cre^−^ Mir137^fl/fl^*	5	7.5	10	7.5	15	10.625

**Table 4 ncrna-10-00001-t004:** Recovered weanlings from *Sox10-cre* litters.

	Male		Female		Total	
	Observed	Expected	Observed	Expected	Observed	Expected
*cre^+^ Mir137^+/+^*	6	5	4	5	10	10.75
*cre^+^ Mir137^fl/+^*	9	9.5	10	9.5	19	21.5
*cre^+^ Mir137^fl/fl^*	6	6.5	7	6.5	13	10.75
*cre^−^ Mir137^+/+^*	6	6.5	7	6.5	13	10.75
*cre^−^ Mir137^fl/+^*	10	10	10	10	20	21.5
*cre^−^ Mir137^fl/fl^*	3	5.5	8	5.5	11	10.75

**Table 5 ncrna-10-00001-t005:** Recovered weanlings from *Baf53b-cre* litters.

	Male		Female		Total	
	Observed	Expected	Observed	Expected	Observed	Expected
*cre^+^ Mir137^+/+^*	9	7	5	7	14	11.25
*cre^+^ Mir137^fl/+^*	10	13	16	13	26	22.5
*cre^+^ Mir137^fl/fl^*	9	5.5	2	5.5	11	11.25
*cre^−^ Mir137^+/+^*	5	5	5	5	10	11.25
*cre^−^ Mir137^fl/+^*	9	8.5	8	8.5	17	22.5
*cre^−^ Mir137^fl/fl^*	3	6	9	6	12	11.25

**Table 6 ncrna-10-00001-t006:** Statistical analysis of recovered weanlings.

Cre Driver Line	Sex-Genotype Independence *p* ^†^	Genotype Effect *p* ^‡^
*Wnt1-cre*	0.6954	0.3535
*Sox10-cre*	0.4832	0.3729
*Nestin-cre*	0.7892	0.9248
*Baf53b-cre*	0.075	0.7376

^†^ Fisher’s exact test; determines if sex and genotype interact. ^‡^ Chi-square test; determines if the distribution of recovered genotypes matches Mendelian ratios.

**Table 7 ncrna-10-00001-t007:** Neuron, glia, and ANNA1/S100β double-positive cell densities and lineage tracing in single *Wnt1-cre* wild-type and conditional knockout bowels.

**Density (/mm^2^)**							
		**Myenteric Plexus**			**Submucosal Plexus**		
		**PSI**	**DSI**	**DC**	**PSI**	**DSI**	**DC**
WT	Neurons	181.9	335.5	148.0	45.65	63.64	11.07
	Glia	343.8	711.8	502.2	260.1	255.2	235.9
	Double-positive	7.609	7.609	2.767	94.77	53.26	1.383
cKO	Neurons	146.6	318.9	298.8	70.56	56.72	22.13
	Glia	388.7	487.7	597.6	310.6	315.4	433.0
	Double-positive	6.917	7.609	2.767	124.5	43.58	4.150
**Lineage Tracing (%)**							
		**Myenteric Plexus**			**Submucosal Plexus**		
		**PSI**	**DSI**	**DC**	**PSI**	**DSI**	**DC**
WT	Neurons	47.15	43.71	29.44	63.64	63.03	62.5
	Glia	100	99.72	98.64	99.90	95.48	96.48
	Double-positive	100	81.82	--	81.75	63.64	--
cKO	Neurons	36.79	31.67	24.07	66.67	92.68	46.88
	Glia	99.82	100	99.54	97.77	99.56	99.36
	Double-positive	80	54.55	--	60	80.95	--

**Table 8 ncrna-10-00001-t008:** Specific miRNAs studied in Hirschsprung disease.

miRNA	Reported Expression	Samples Compared	References	Present in smRNA-seq	ENCDC-Enriched
miR-124	↑	Stenotic vs. normal colon	[47]	Y	Y
miR-181a	↑	Case vs. control colon	[60]	Y	Y
miR-218	↑	Case vs. control colon	[61,62]	Y	Y
miR-24	↑	Case vs. control colon	[63]	Y	Y
miR-488	↑	Case vs. control colon	[50]	Y	Y
miR-92a	↑	Case vs. control colon	[64]	Y	Y
let-7a	↑	Case vs. control colon	[63]	Y	N
miR-1251	↓	Aganglionic vs. control colon	[65]	Y	N
miR-128	↑	Case vs. control colon	[66]	Y	N
miR-1324	↑	Aganglionic vs. control colon	[67]	N	N
miR-140	↓	Stenotic vs. dilated colon	[68]	Y	N
miR-141	↑/↓	Case vs. control colon	[69]/[70]	Y	N
miR-142	↓	Case vs. control samples	[71]	Y	N
miR-144	↓	Case vs. control colon	[72]	Y	N
miR-146a	↑	Case vs. control samples	[73]	Y	N
miR-146b	↑	Case vs. control colon	[74]	Y	N
miR-148a	↑	Aganglionic vs. ganglionic colon	[75]	Y	N
miR-150	↑	Case vs. control colon	[76]	Y	N
miR-192	↑/↓	Aganglionic vs. ganglionic colon/Case vs. control colon	[77]	Y	N
miR-195	↑	Case vs. control colon, aganglionic vs. control colon	[78,79,80]	Y	N
miR-199a	↑	Case vs. control plasma exosomes and colon	[81]	Y	N
miR-200a	↑/↓	Aganglionic vs. ganglionic colon/Case vs. control colon	[77]/[82]	Y	N
miR-200b	↑	Aganglionic vs. ganglionic colon	[77]	Y	N
miR-206	↓	Case vs. control colon, aganglionic vs. control colon	[83,84]	Y	N
miR-214	↑	Case vs. control colon	[85]	Y	N
miR-215	↓	Case vs. control colon	[86]	Y	N
miR-30a	↓	Case vs. control colon	[69]	Y	N
miR-31/-31*	↓	Case vs. control colon	[87]	Y	N
miR-369	↑	Aganglionic vs. control colon	[88]	Y	N
miR-424	↑	Dilated vs. control colon	[89]	N	N
miR-431	↑	Case vs. control colon	[90]	Y	N
miR-4516	↑	Dilated vs. control colon	[91]	N	N
miR-483	↑/↓	Stenotic vs. dilated colon/aganglionic vs. normal colon	[92]/[93]	Y	N
miR-637	↑	Aganglionic vs. control colon	[94]	N	N
miR-770	↓	Case vs. control colon	[95]	Y	N
miR-938	↑	Risk vs. non-risk fibroblasts, tibial nerve	[96]	N	N
miR-939	↑	Case vs. control colon	[97]	N	N

## Data Availability

The data presented in this study can be accessed under GEO accession number GSE250528.

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
