# Peer review of "Sequencing Reveals miRNAs Enriched in the Developing Mouse Enteric Nervous System"

_ncrna, 2023, doi:10.3390/ncrna10010001_

Round 1

Reviewer 1 Report

Comments and Suggestions for Authors

Authors reported a RNA sequencing study to investigate the miRNAs differentially expressed in the enteric nervous system (ENS) cells compared to the gut during ENS development using mouse model. They identified 73 differentially expressed miRNAs and selected miRNA-137 among the top six candidate miRNAs for functional follow-up. Conditional knockout of Mir137 at different stages of ENS development across four Cre driver lines did not show significant differences in survival rate nor count of neurons/glia of the knockout mice. Overall the study is carefully carried out but lacks mechanistic insights into the gene regulatory function of the top and selected miRNAs. 

Major comments:

1) Although the GO analysis was performed on mRNAs predicted as targets of the selected top miRNAs, no comprehensive integrative analysis between the mRNA and miRNAs was performed to investigate the possible mechanisms of dysregulation. While no obvious change in ENS composition was observed in the conditional knockout models, it is still worth investigating the possible disruption in gene expression of target genes in ENS and surrounding gut cells due to the loss of the selected miRNAs.

2) No hypothesis on the possible molecular mechanisms of the miRNA in the ENS development was given. Are the top differentially expressed miRNA mainly targeted genes expressed in ENS or the surrounding gut cells? Any evidence to suggest the involvement of non-ENS cells in miRNA-mediated gene regulation in ENS development?

3) It was mentioned that disruption of miRNA biogenesis via deletion of Dicer in mouse model may result in loss of ENCDCs by E17. Could the loss of ENCDCs be the consequence of disruption of a combination of miRNAs instead of having involvement of a few major miRNAs? Looking into the overlapping gene regulatory network disrupted by the top miRNAs may give us a clue on the possible interaction.

Author Response

Dear Reviewer 1,

Thank you for your thorough review of our work! Below is our response to your points.

1) We recognize that our paper lacks detailed information about potential gene-level effects of miR-137 deletion. Prior to beginning experiments, we attempted to synthesize our mRNA and miRNA data more directly, but failed to find anything informative. Based on this point, we decided to incorporate reporting this process and its results into our revisions.

2) The genes evaluated for the GO analyses were solely those expressed in the ENS, and miR-137 is almost solely expressed in the ENS. That said, we have added discussion of the known crosstalk between ENCDCs and surrounding cells as part of ENS development.

3) We agree that it is highly possible that ENS development depends on multiple miRNA, rather than one single miRNA. While we do not have access to the tools that would allow us to evaluate all of the overlapping gene networks as suggested, we have generated a Venn diagram of the top miRNAs’ target counts as a supplemental figure (Supplemental Figure 1) and added discussion of the 3 genes targeted by all 5 miRNAs.

Reviewer 2 Report

Comments and Suggestions for Authors

Overall, this is a very nicely done and well written publication. The study design is appropriate and apparently, the analyses were carefully performed. This manuscript shows rich and valuable content, which is within the journal’s scope. However, the study contains minor points which need to be clarified before publication.

My comments:

Line 28 - From anatomical point of view gastrointestinal tract (also known as alimentary tract or digestive tract) is a tract which passes food from mouth to the anus. ENS is present not in whole GIT!

Line 54 – please clarify the hypothesis and aims of this study

Line 64 - number of the approval number of Ethic Committee is missing.

Line 135 - how the authors were able to precisely localize “proximal and distal small intestine”? What was reference point. Moreover, terms “proximal and distal small intestine” are anatomically incorrect. In animals, small intestine can be divided into duodenum, jejunum and ileum but not to “distal and proximal”. It raises a question what in fact, the authors dissected out.

Line 135 – why only one segment of large intestine was studied (what about cecum and rectum)?

Line 138 – as previously. Division into “proximal and distal colon” is anatomically incorrect (colon is divided into ascending, transverse and descending parts).

Line 141 – change to “4oC”

Line 143 - From histological point of view there are only four kinds of tissues: epithelial, connective, muscular and nervous. Therefore, such terms as “bowel tissue” “colon tissue ““stenotic tissue” (table 8), are not justified. The authors terribly confuse organs with tissues. Organs are assembled from the four basic types of tissues and have cells with specialized functions.

Line 145 – did the authors somehow check the specificities of primary antisera used. Please describe how.

Figure 2, 3 – images are too small to make any judgement.

Please add conclusions chapter at the end.  

Author Response

Dear Reviewer 2,

Thank you for your detailed review of our work! We appreciate your attention to details and precision in language.

While we agree that we would ideally refer to the intestinal segments studied as duodenum, ileum, descending colon, etc, we unfortunately lose a lot of information about anatomical positioning during bowel dissection, straightening, and (most impactfully) stretching. We added commentary on what portions of bowel we are likely examining to the Methods section, but chose to retain the original labels for our data and those reported in prior literature in the name of open communication about what we do and do not know about our samples.

In response to two of your specific questions:

> Line 135 – why only one segment of large intestine was studied (what about cecum and rectum)?

While there are many bowel regions that could be studied, we decided to examine the distal colon because this region is aganglionic (completely lacking ENS) in people with Hirschsprung disease. Furthermore, the end of the colon is commonly affected by mutations that reduce proliferation, induce premature differentiation, or impair migration of ENS precursors during fetal development. Both the seemingly normal ENS in all bowel regions we examined and the grossly typical growth of cKO mice suggest that there is little utility to evaluating ENS in other bowel regions. Cecum is rarely examined in mouse ENS studies, as its anatomy and function in mice are difficult to compare to human bowel anatomy.

> Line 145 – did the authors somehow check the specificities of primary antisera used. Please describe how.

While we have used these ANNA1 and S100β antibodies in previous reports, we understand that precedence is insufficient to justify trusting any given antibody. That said, our staining protocol indicates they are functioning as expected based on their co-labeling of Wnt1-cre lineage traced cells, the morphology and localization of staining, and visual consistency between our acquired images and those published internally and elsewhere in literature by other labs (including ANNA1+/S100β+ cells). This information has been added to our Methods section.

We have otherwise updated our terminology and corrected our errors according to your remarks. To assist in evaluating the images from figures 2 and 3, we have also produced supplementary figures containing solely the merged images from both figures for ease of reading (Supplemental Figure 2, 3).

Reviewer 3 Report

Comments and Suggestions for Authors

The manuscript by Pai et al. presents a comprehensive analysis of miRNAs within the context of the developing enteric nervous system (ENS). The research methodology is robust, and the results are meticulously documented. The authors have adeptly addressed and rationalized the deviations from previously published findings in this domain. Although miR-137 appears to be non-contributory in the development of the ENS, the identification of other differentially expressed miRNAs lays a solid groundwork for future research. These findings are particularly significant for elucidating the role of miRNAs in the ontogeny of the ENS and their potential implications in enteric neuropathologies such as ENCR. I find no areas necessitating further refinement in this manuscript and thus endorse its publication.

Author Response

Dear Reviewer 3,

Thank you for your glowing review of our work! We are delighted that you find this paper acceptable as-is.